# Effects of partial penectomy for penile cancer on sexual function: A systematic review

Eleanor Whyte[1], Alexandra Sutcliffe[1], Philip Keegan[1], Tom Clifford[2], Jamie Matu[3], Oliver M. Shannon[4], Alex Griffiths [3]*

1 South Tyneside and Sunderland NHS Foundation Trust, Sunderland Royal Hospital, Sunderland, United Kingdom, 2 School of Sport, Exercise and Health Science, Loughborough University, Loughborough, United Kingdom, 3 School of Health, Leeds Beckett University, Leeds, United Kingdom, 4 Population Health Sciences Institute, Newcastle University, Newcastle Upon Tyne, United Kingdom

* A.Griffiths@LeedsBeckett.ac.uk

**Data Availability Statement:** All relevant data are within the manuscript and its Supporting Information files.

**Funding:** The authors received no specific funding for this work.

## Abstract

Penile cancer is a rare but debilitating condition, which often requires aggressive treatment. Partial penectomy is considered as a treatment option when a sufficient portion of the penile shaft can be maintained to preserve functionality. This systematic review, which followed the PRIMSA guidelines, aimed to evaluate the effects of partial penectomy for penile cancer on sexual function—the maintenance of which is often a priority in patient groups—and to identify potential factors which may moderate these effects. A systematic search of PubMed, The Cochrane Library, and Open Grey as well as MEDLINE, CINAHL and Open Dissertations via EBSCOhost was conducted from inception through to 24th March, 2022. Studies were required to include adults aged ≥18 years who had undergone partial penectomy for the treatment of penile cancer, with a quantitative measure of sexual function available pre- and post-surgery. Four eligible articles were identified for inclusion in this review, three of which reported a decrease in sexual function pre- to post-surgery across all domains of the International Index of Erectile Function (IIEF) questionnaire (erectile function, orgasmic function, sexual desire, intercourse satisfaction and overall satisfaction). Conversely, one study reported an increase in sexual function across IIEF domains, except for orgasmic function, which decreased, pre- to post-surgery. Greater penile length was associated with higher post-operative sexual function, whilst increasing age and higher anxiety levels were associated with lower post-operative sexual function levels in one study. Despite the overall drop in sexual function, many patients were still able to maintain satisfactory sex lives following partial penectomy. Given the limited research in this area and small sample sizes across studies, additional well-controlled investigations are warranted to provide further evidence on the effects of partial penectomy for penile cancer on sexual function.

## Introduction

Penile cancer is a rare malignancy in the Western world, with an estimated annual incidence of <1 per 100,000 individuals in Europe and the United States [1, 2]. In contrast, penile cancer rates are much higher in the developing world, where this condition represents a greater public

**Competing interests:** The authors have declared that no competing interests exist.

health concern [3]. For example, in Brazil and Uganda, annual penile cancer incidence rates are approximately 6–8 [3] and 4 [4] per 100,000 individuals, respectively. Meanwhile, in India, penile cancer is one of the most common genitourinary cancers and has been estimated to account for up to 6% of all cancer cases in men [3, 5]. Risk factors for penile cancer are varied and, in some cases, modifiable, and include smoking, lack of neonatal circumcision, chronic inflammation, history of phimosis, number of sexual partners, and infection with human papillomavirus (HPV) [3, 6].

The most common form of penile cancer is squamous cell carcinoma, which accounts for 70–75% of all cases and is characterised by early metastatic spread [3]. Other, less common sub-types of penile cancer include basaloid, sarcomatoid and warty subtypes, which are aggressive, and the verrucous and condylomatous sub-types, which rarely spread and have much better prognosis [3]. The traditional surgical treatment option for penile cancer is radical (i.e., total) penectomy with perineal urethrostomy, which results in an inability to engage in penetrative sexual intercourse and void in the upright position. More conservative approaches (so called 'organ sparing surgery') have become popular in recent years, which show higher risk of local recurrence compared with radical penectomy, but do not appear to impact cancer-specific survival rates [7]. Nevertheless, in cases where penile cancer is more advanced, partial or radial penectomy are the current therapeutic options [8, 9].

Partial penectomy is recognised as an effective treatment option for penile cancer with low recurrence rates [9]. This surgical procedure is considered when a sufficient portion of the penile shaft can be preserved to enable functionality (e.g., direction of the urinary stream). Nevertheless, and perhaps unsurprisingly, partial penectomy can have a deleterious effect on sexual function, the maintenance of which is often a priority in patient groups [10]. For example, Sansalone et al. [9] reported decrements in erectile dysfunction, orgasmic function, sexual desire, intercourse satisfaction and overall satisfaction following partial penectomy. However, encouragingly, these decrements were typically small-to-moderate in magnitude such that many patients were able to maintain sexual outcome levels only marginally lower than those pre-surgery. In another similar study by Yu et al. [11], patients reported significantly reduced sexual function following partial penectomy. Interestingly, however, those authors provide evidence to suggest that the response to partial penectomy could differ notably depending upon participant characteristics such as age (negatively associated with sexual function) and penile length (positively associated with sexual function).

A better understanding of the effects of partial penectomy on sexual function, alongside factors that could explain a differential response to this procedure between patients, would be valuable to inform clinicians and help manage patient expectations. Therefore, in the current study we aimed to conduct a systematic review of the extant literature exploring effects of partial penectomy on sexual function. This research will: i) provide clarity on the effects of partial penectomy for penile cancer on sexual function, ii) identify potential effect moderators, and iii) highlight potential gaps in the literature, to help inform the direction of future research.

## Methods

The current systematic review followed the Preferred Reporting Items for Systematic Review and Meta-analyses (PRISMA) guidelines (38), and was prospectively registered on the PROSPERO database (CRD42021250248).

### Literature search

A systematic search of PubMed, The Cochrane Library, and Open Grey as well as MEDLINE, CINAHL and Open Dissertations via EBSCOhost was conducted from inception through to

11<sup>th</sup> May, 2021, and updated on 24<sup>th</sup> March, 2022. Searches were conducted using pre-defined search terms relating to partial penectomy and sexual function, with Boolean operators and MeSH terms utilised where appropriate. No publication date or language restrictions were applied. The search strategy, which was devised by an information specialist who is experienced with systematic reviews (JM), was tailored to the requirements of each database (**S1 File**). In addition, a manual search of the reference lists of eligible studies and recent review articles was also performed to identify further relevant research.

## Study selection

Inclusion criteria were based on the following Population, Intervention, Comparator, Outcome, Study design (PICOS) criteria: *Population*: Studies involving adults aged $\geq$18 years (no exclusion criteria were applied for smoking history or health status); *Intervention*: Studies in which patients have undergone partial penectomy for the treatment of penile cancer; *Comparator*: Studies which report sexual function before (comparator) and after (outcome) partial penectomy. Given the nature of the studies, no separate control group was required; *Outcome*: Studies which provide quantitative measures of sexual function using a validated screening questionnaire. Information on potential effect moderators was extracted where present, although the lack of data on effect moderators was not considered an exclusion criteria; *Study design*: Primary research studies (no further exclusion criteria were applied in relation to study design). In addition, only studies published in the English language were considered eligible for this review.

## Screening

Two researchers (EW and AS) independently screened the titles and abstracts of retrieved papers to evaluate their eligibility for inclusion. Potentially eligible articles were moved to the next stage (i.e., full-text appraisal), whilst ineligible articles were excluded. The same two researchers (EW and AS) independently appraised the full-texts of the selected articles. Disagreements between the researchers at all stages of the review were resolved by consultation with a third reviewer (OMS).

## Data extraction

Data from eligible full texts was extracted by one reviewer (EW) and checked by a second reviewer (AS). A third member of the research team (OMS) was available to resolve any conflicts. An electronic form which has been used previously by members of the research team [12, 13] was adapted for this purpose. The following information was extracted: Surname of the first author, publication year, country, date of surgery, follow up duration, sample size, participant age, body mass index (BMI), presence of other comorbidities, history of previous penile surgery, relationship status, pre- and post-surgery penile characteristics, tumour characteristics (e.g., tumour size, stage, histological type), details of surgical procedure performed, surgical complications, quantitative measures of sexual function, information on statistical analysis.

## Data synthesis

Data were deemed unsuitable for meta-analysis due to the small number of eligible papers [14] and single arm pre-test post-test study designs employed [15]. Therefore, a narrative (descriptive) synthesis of the literature was conducted. Key findings were tabulated and explored qualitatively in the text. Potential effect moderators were highlighted and discussed.

## Assessment of study quality

Risk of bias of the included studies was assessed by one researcher (AG) using the ROBINS-I tool for non-randomised studies of interventions [16]. Results were checked for accuracy by a second researcher (OMS).

# Results

## Search results

A total of 279 articles were identified as part of the database searches. Following the removal of duplicates, 141 titles and abstracts were screened, and 27 full-text articles were retrieved for further appraisal. Evaluation of the full-text studies identified 4 articles eligible for inclusion in this systematic review [9, 11, 17, 18]. Most full texts appraised were deemed ineligible for inclusion as they did not report pre-operative sexual function [10, 19–31]. Additional reasons for exclusion include studies reporting the wrong outcome measures [32–37] and the full text being unavailable in English [38–40]. A summary of the screening process is provided in **Fig 1**.

## Study characteristics

The characteristics of included studies are presented in **Table 1**. The total number of patients from the 4 eligible articles was 94, with the sample size of included studies ranging from 8 to 43). The median participant age was 59 years, and ranged from 25 to 86 years. All of the studies reported pre- and post-operative sexual function via the International Index of Erectile Function (IIEF) [41], whilst Wan et al. [18] also reported pre- and post-operative values for the Self-Esteem and Relationship (SEAR) questionnaire [42]. Measures of sexual function were reported prospectively for all of the studies except for Romero et al. [17] who asked patients to rate their pre-operative sexual function retrospectively.

## Risk of bias

Overall, there was a serious risk of bias in all included studies (**Table 2**). The risk of bias due to confounding was appraised as serious in all studies. Bias in selection of participants into the study was moderate in all studies. In contrast, bias in classification of interventions and bias due to missing data was low in all studies. Risk of bias in measurement of the outcome was mixed, with one study demonstrating serious risk [17], another study moderate risk [9], and two studies low risk [11, 18]. In all studies, there was insufficient evidence to appraise the risk of bias in selection of the reported results. Further, bias due to deviations from intended interventions was not relevant to the study design of included studies.

## Effects of partial penectomy on sexual function

**International index of erectile function domains.** The IIEF questionnaire reports sexual function across 5 separate domains, including erectile function, orgasmic function, sexual desire, intercourse satisfaction and overall satisfaction. Compared with pre-operative values, three studies reported a significant decrease in all individual IIEF domains following partial penectomy [9, 11, 17]. In contrast, one study reported a significant increase in IIEF domains, except for orgasmic function which decreased, following partial penectomy [18] (**Table 3**). Further details of the effects of partial penectomy on each IIEF domain are provided below:

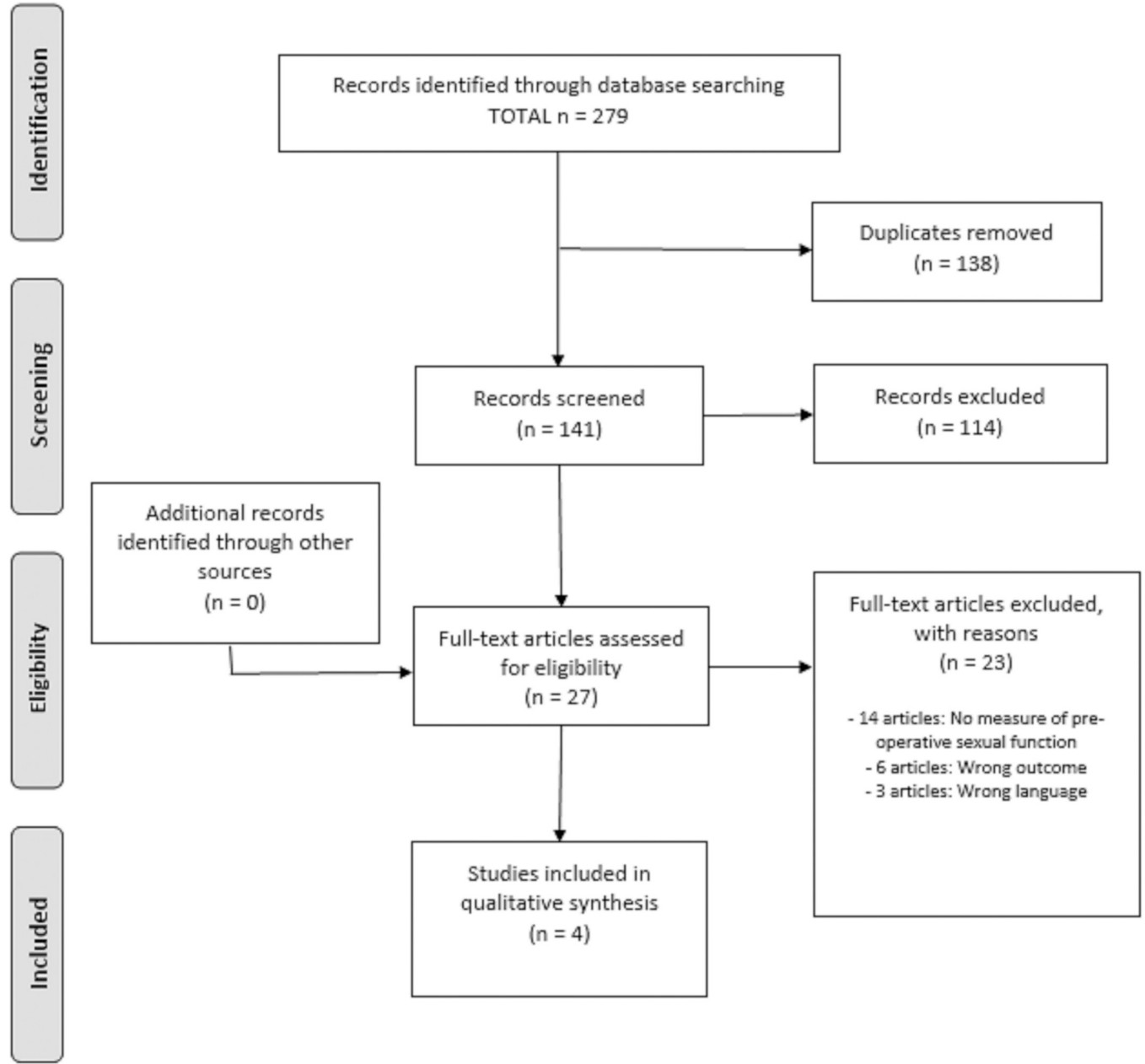

**Fig 1. PRISMA flow chart of the study selection process.**

### Erectile function

Three studies reported a decrease, whilst one study reported an increase, in erectile function following partial penectomy. Specifically, in the study by Romero et al. [17], erectile function scores decreased by an average of 34.4% pre- to post-surgery. Ten out of 18 patients reported erection of the penile stump hard enough for penetration 'always' or 'most times' (i.e., more than half the time) following surgery, which was similar to pre-surgery levels. Two patients reported a reduction in erectile function from 'always' to 'sometimes' and 'almost never', whilst six patients reported having 'no sexual activity'. Sansalone et al. [9] reported a similar

**Table 1. Characteristics of studies exploring effects of partial penectomy for penile cancer on sexual function.**

| Study | Country | Time of pre-operative sexual function measure | Time of Post-operative sexual function measure | Sample size (n) | Age (years) | Relationship information | Post-surgery penile characteristics | Tumour characteristics | Description of surgical procedure |
|---|---|---|---|---|---|---|---|---|---|
| Romero et al. [17] | Brazil | Retrospective recall median 23.5 months post-operation | 23.5 months post-operation (range: 6–62 months) | 18 | Median: 52 (range: 35–86) | 14 patients had a steady partner relationship | Median: 4 cm | All patients had squamous cell carcinoma. | Partial penectomy with a 2-cm margin of tumour-free tissue |
| | | | | | | | | T1: n = 12 | |
| | | | | | | | | T2: n = 2 | |
| | | | | | | | | T3: n = 4 | |
| | | | | | | | | Average tumour size: 3.4 cm. | |
| Sansalone et al. [9] | Italy | Pre-operation (time not specified) | 3 months post-operation | 25 | Mean and SD: 61.5 ± 2.5 (range: 25–75) | 24 patients were married. All were sexually active | All ≥ 3 cm, Range: 3–4.5 cm | Histological type not specified. | Organ sparing partial penectomy with pseudoglans reconstruction |
| | | | | | | | | T1a: n = 6 | |
| | | | | | | | | T1b: n = 5 | |
| | | | | | | | | T2: n = 14 | |
| Wan et al. [18] | China | 4 weeks pre-operation | 6-months post-operation | 8 | Mean and SD: 62.0 ± 9.8 (range: 44–74) | All patients were married and sexually active | All ≥3 cm | All were squamous cell carcinoma. | Partial penectomy with margin negativity ascertained during operation to maximise penile length |
| | | | | | | | | T1: n = 5 | |
| | | | | | | | | T2: n = 3 | |
| Yu et al. [11] | China | Pre-operation (time not specified) | 6-months post-operation | 43 | Median: 56 (range/ SD not reported) | 34 patients reported having a partner. All reported regular sexual activity. | Median: 4 cm | Histological type not specified. | Partial penectomy according to approved guidelines |
| | | | | | | | | Grade not specified. | |

decrease in erectile function score by 25.8% pre- to post-surgery. Seventeen out of 25 patients, reported erection of the penile stump hard enough for penetration 'always' or 'most times' following surgery, which was similar to pre-surgery levels. In contrast, five patients reported moderate erectile dysfunction. Similarly, Yu et al. [11] reported an overall decrease in erectile function score pre- to post-surgery by 33.3%. Twenty one out of 43 patients reported erectile function that 'always' or 'most times' allowed for sexual intercourse following surgery, whilst 12 patients reported an erection hard enough for penetration 'sometimes' or 'a few times', and

**Table 2. ROBINS-I quality assessment of included studies.**

| Study | Domain 1: Bias due to confounding | Domain 2: Bias in selection of patients into the study | Domain 3: Bias in classification of interventions | Domain 4: Bias due to deviations from intended interventions | Domain 5: Bias due to missing data | Domain 6: Bias in measurement of the outcome | Domain 7: Bias in selection of the reported results | Overall |
|---|---|---|---|---|---|---|---|---|
| Romero et al. [17] | Serious | Moderate | Low | N/A | Low | Serious | No information | Serious |
| Sansalone et al. [9] | Serious | Moderate | Low | N/A | Low | Moderate | No information | Serious |
| Wan et al. [18] | Serious | Moderate | Low | N/A | Low | Low | No information | Serious |
| Yu et al. [11] | Serious | Moderate | Low | N/A | Low | Low | No information | Serious |

N/A = not relevant to non-randomised pre-post studies

**Table 3. Pre- and post-operative measures of sexual function across the included studies as determined by the International Index of Erectile Function (IIEF) questionnaire.**

| Study | IIEF domain | | | | | | | | | |
|---|---|---|---|---|---|---|---|---|---|---|
| | Erectile function | | Orgasmic function | | Sexual desire | | Intercourse satisfaction | | Overall satisfaction | |
| | Pre | Post | Pre | Post | Pre | Post | Pre | Post | Pre | Post |
| Romero et al. [17] | 29.56 ± 1.42 | 19.39 ± 12.44* | 9.94 ± 0.24 | 7.67 ± 3.90* | 8.89 ± 0.76 | 7.61 ± 1.94* | 12.67 ± 1.46 | 6.89 ± 5.57* | 8.61 ± 1.58 | 6.11 ± 2.65* |
| Sansalone et al. [9] | 28.68 ± 1.04 | 21.28 ± 3.07* | 9.86 ± 0.59 | 7.92 ± 0.86* | 8.75 ± 1.67 | 7.16 ± 0.94* | 12.50 ± 1.75 | 7.32 ± 2.65* | 9.01 ± 0.79 | 6.52 ±1.84* |
| Wan et al. [18] | 11.75 ± 1.83 | 20.38 ± 2.26* | 3.75 ± 1.75 | 2.13 ± 0.64* | 2.75 ± 0.89 | 6.00 ± 1.31* | 2.63 ± 0.74 | 5.50 ± 1.41* | 2.63 ± 0.74 | 6.75 ± 1.67* |
| Yu et al. [11] | 26.70 ± 3.07 | 17.81 ± 10.66* | 8.44 ± 1.16 | 5.81 ± 3.35* | 8.33 ± 1.27 | 6.28 ± 2.16* | 12.30 ± 2.21 | 7.07 ± 4.56* | 8.00 ± 1.19 | 5.91 ± 2.01* |

N.B. Data for Romero et al. were reported as median ± SD. Data for all other studies is reported as Mean ± SD. Values are expressed to 2 DP.

* = significantly different to pre-surgery values. The pre- and post-operative values for orgasmic function in the study by Wan et al. were reported in the manuscript as 2.13 ± 0.64 and 3.75 ± 1.75 pre- and post-operative, respectively. There was inconsistency between the numerical results reported in the table and descriptive text provided in the manuscript. Therefore, the authors were contacted for clarification on the direction of change. The authors confirmed that orgasmic function decreased post-surgery, and these values have therefore been reversed.

10 patients reported 'no sexual activity' or 'almost never' engaging in sexual activity. Finally, in contrast to the other studies, Wan et al. [18] reported a notable increase in erectile function scores by 73.4% post-partial penectomy, although those authors did not provide a breakdown of erectile function at the individual participant level.

## Orgasmic function

All four identified studies reported a decrease in orgasmic function following partial penectomy. Specifically, in the study by Romero et al. [17], orgasmic function scores decreased by an average of 22.8% pre- to post-surgery. Thirteen out of 18 patients reported that they ejaculated and experienced orgasm 'always' or 'almost always' when they had sexual stimulation or intercourse following partial penectomy. Meanwhile, two patients reported ejaculation and orgasm 'sometimes' or 'a few times' and, three patients reported no orgasmic function post-surgery compared with 'always' or 'almost always' pre-surgery. A similar decrease (19.7%) in orgasmic function scores were reported pre- to post-surgery in the study by Sansalone et al. [9]. Sixteen out of 25 patients reported that they ejaculated and had the feeling of orgasm 'always' or 'almost always' when they had sexual stimulation or intercourse following partial penectomy, whilst 3 patients did not reach orgasm. Likewise, in the study by Yu et al. [11], orgasmic function decreased by 31.2% pre- to post-surgery. Twenty eight out of 43 patients reported that they ejaculated and had the feeling of orgasm 'always' or 'most times' during sexual stimulation or intercourse following partial penectomy. In contrast, 5 patients reported ejaculation or orgasm 'sometimes' or 'a few times', and 10 patients reported having 'no intercourse' or 'almost never' ejaculating/ reaching orgasm. Finally, Wan et al. [18] reported a 43.2% decrease in orgasmic function scores pre- to post-surgery, although a breakdown of orgasmic function at the individual participant level was not provided.

## Sexual desire

Three studies reported a decrease, whilst one study reported an increase, in sexual desire following partial penectomy. Specifically, Romero et al. [17] reported a significant decrease in sexual desire by an average of 14.4% pre- to post-surgery. Eight out of 18 patients reported 'high' or 'very high' sexual desire 'always' or 'most times' before and after surgery. Four patients reported the same 'moderate' level of sexual desire pre- and post-surgery, whilst 6 patients reported a reduction in the level ('moderate' to 'low') and/or frequency ('a few times'

to 'sometimes') of sexual desire following surgery. Meanwhile, in the study by Sansalone et al. [9], who reported a decrease in sexual desire scores by an average of 18.2% pre- to post-surgery, 14 out of 25 patients reported sexual desire 'always' or 'most times', whilst five patients reported a reduction in the level ('moderate' to 'low') and/or frequency ('a few times' to 'never') of sexual desire following surgery. Yu et al. [11] also reported a decrease in sexual desire scores, which dropped by 24.6% pre- to post-surgery. Those authors found that 26 out of 43 patients had 'high' or 'very high' sexual desire 'always' or 'most times' following partial penectomy, 12 patients reported sexual desire 'sometimes' or 'a few times', and 5 patients reported 'almost never' feeling sexual desire. Finally, in contrast to the other investigations, Wan et al. [18] reported a notable increase in sexual desire scores by 118.2% pre- to post-surgery.

## Intercourse satisfaction

Intercourse satisfaction was decreased in three studies, and increased in one study, following partial penectomy. Specifically, in the study by Romero et al. [17], intercourse satisfaction scores decreased by 45.6% pre- to post-surgery. Sexual frequency was the same pre- to post-operation in six out of 18 patients, with three, two, and one of the patients reporting a sexual frequency of 7 to 10, 5 to 6, and 3 to 4 times, respectively, over a four-week period. The other 12 patients reported a reduced sexual frequency, six of whom reported no intercourse. Ten patients reported that, when they engaged in sexual intercourse, it was 'always' or 'almost always' satisfactory, whilst two patients reported that intercourse was satisfactory only 'a few times'. Three patients reported that their post-operative sexual intercourse was 'highly enjoyable' or 'very highly enjoyable', similar to pre-operative levels, whilst five patients reported that their sexual intercourse was maintained as 'fairly enjoyable' pre- to post-surgery, and four reported a decrease in satisfaction to 'not very enjoyable' or 'fairly enjoyable'. In the study by Sansalone et al. [9], intercourse satisfaction scores decreased by 41.4% pre- to post-surgery. Seven of the 25 patients maintained the same sexual frequency as pre-operation. However, the majority reported a reduction in sexual frequency and two patients did not attempt intercourse post-surgery. Sexual intercourse and satisfaction varied between patients, although the majority reported finding intercourse 'almost always' or 'a few times' satisfactory, and their satisfaction was rated as 'fairly' or 'highly enjoyable'. Yu et al. [11] reported a decrease in intercourse satisfaction scores by 42.5% pre- to post-surgery. Of the 43 patients, 12 patients reported more than seven attempts at sexual intercourse, whilst 19 patients reported 3–6 attempts at sexual intercourse over a 4-week period post-surgery. Sixteen patients reported that their sexual intercourse was 'always' or 'most times' satisfying, 16 patients reported feeling satisfied 'sometimes' or 'a few times', and 11 patients 'never' felt satisfied or did 'not attempt intercourse'. When rating their enjoyment, 10 patients reported that their sexual intercourse was 'highly' or 'very highly' enjoyable, 19 patients rated their intercourse as 'fairly enjoyable', and 14 patients reported that their sexual intercourse had 'no enjoyment' or else they reported 'no intercourse'. Finally, Wan et al. [18] reported an increase in intercourse satisfaction scores 109.1% pre- to post-surgery.

## Overall satisfaction

Overall satisfaction was decreased in three studies, and increased in one study, pre- to post-surgery. Specifically, Romero et al. [17] reported a significant decrease in overall satisfaction scores by 29.0% post-partial penectomy. Prior to surgery, all eighteen patients reported being 'moderately' or 'very satisfied' with their overall sex life and sexual relationship with their partners. However, post-surgery, only six individuals maintained a similar degree of satisfaction to

pre-surgery levels. Five patients reported being 'equally satisfied and dissatisfied', four reported being 'moderately dissatisfied' and three were 'very dissatisfied'. Sansalone et al. [9] reported a similar reduction in overall satisfaction by 27.6% pre- to post-surgery. Of the 25 patients, seven reported being 'very satisfied' with their overall sex life and sexual relationship with their partners, whilst two reported being 'very dissatisfied' after partial penectomy. The remaining patients reported that they were 'equally satisfied and dissatisfied'. Yu et al. [11] reported a 26.1% decrease in overall satisfaction scores pre- to post-surgery. Of the 43 patients in this study, seven reported being 'very' or 'moderately satisfied', 28 patients being 'equally satisfied and dissatisfied', and eight patients reported being 'moderately dissatisfied' with their overall sex life and sexual relationship with their partners after surgery. Finally, Wan et al. [18] reported a 156.7% increase in overall satisfaction pre- to post-surgery.

## Effect moderators

One study was identified that explored potential factors which may moderate the effects of partial penectomy on sexual function as measured by the IIEF questionnaire. Yu et al. [11] reported that post-operative penile length was significantly associated with higher intercourse satisfaction scores in univariate analyses. In contrast, greater age and self-rating anxiety score were associated with worse sexual function outcomes across all IIEF domains [11].

## Self-esteem and relationship questionnaire

Wan et al. [18] also measured the effects of partial penectomy on the self-esteem and sexual relationships of the patients via the SEAR questionnaire, which includes sub-domains focused on sexual relationships, self-esteem and overall relationships [42]. Scores across all three domains were significantly increased in patients following partial penectomy, with higher scores reflecting better response to treatment. Sexual relationship scores increased from 47.5 ± 10.18 to 75.31 ± 11.05 pre- to post-surgery, whilst self-esteem and overall relationship scores increased from 45.63 ± 13.74 to 68.75 ± 16.42 and 41.25 ± 11.26 to 71.25 ± 11.26, respectively.

## Discussion

This study aimed to systematically review the effects of partial penectomy for penile cancer on measures of sexual function, and to identify potential factors which might moderate these effects. Three out of four identified papers reported a significant decrease in all IIEF sexual function domains, whilst one study reported an increase in sexual function across IIEF domains (except for orgasmic function, which decreased), following partial penectomy. Promisingly, many of the patients across studies were still able to maintain satisfactory sex lives post-surgery, with around half reporting that they were able to maintain erection of the penile stump hard enough for penetration 'always' or 'most times'. A similar number reported ejaculation and the experience of an orgasm 'always' or 'almost always' when they had sexual stimulation or intercourse following partial penectomy.

The findings of Yu et al. [11] suggest that maintaining greater penile lengths may help to improve intercourse satisfaction. Therefore, surgical approaches which help spare penile length may be encouraged (where clinically appropriate) for maintaining post-operative sexual function. Both Romero et al. [17] and Yu et al. [11] reported an incision site which allowed a 2-cm margin of tumour-free tissue. However, more recent research suggests that safe surgical margins can be reduced from 2 cm to 3–5 mm, thus allowing greater preservation of penile lengths [43]. As such, patients undergoing partial penectomy may now expect to maintain greater penile lengths and sexual function than individuals historically undergoing this

operation. Yu et al. [11] also noted that greater age was associated with significantly worse sexual function post-partial penectomy, as were higher levels of anxiety according to Zung's Self-Rating Anxiety Scale (SAS) [44]. This suggests that multidisciplinary follow up with a psychologist trained in sexual therapy, who could help address raised anxiety levels alongside other psychological consequences of the surgery, may be beneficial to aid recovery of post-operative sexual function. This may be especially important in younger individuals who typically self-report a higher sexual frequency compared with older adults [45], and who may not have yet completed their family.

Although Romero et al. [17], Sansalone et al. [9], and Yu et al. [11] all reported a decrease in sexual function (across all IIEF domains) following partial penectomy, it is interesting to note that Wan et al. [18] actually reported an increase in all IIEF domains (except for orgasmic function, which decreased) post-surgery. It is possible that these conflicting findings could be related to differences in the surgical technique employed between studies. For example, Wan et al. [18] reported collecting intraoperative frozen sections during surgery to ascertain margin negativity, which allowed maximal preservation of penile lengths. Whilst post-operative penile lengths appeared to be similar to those reported in other studies (i.e., 3–4 cm), it is possible that this technique allowed a relatively higher proportion (i.e., % of original length) of the penis to be preserved compared with other investigations, although this cannot be confirmed based around the available data. Alternatively, pre-operative sexual function scores across all IIEF domains were typically lower in the study by Wan et al. [18], such that there may have been more scope for improvement of sexual function scores with surgery. Despite the lower pre-operative values reported by Wan et al. [18], post-operative scores across IIEF domains were similar to those reported in other studies. Wan et al. [18] also reported pre- and post-operative sexual function scores according to the SEAR questionnaire. Although not measured pre-operatively in any of the other investigations, Sansalone et al. [9] administered the SEAR questionnaire post-operatively only and again reported similar post-operative scores to those of Wan et al. [18]. This suggests that despite the different surgical techniques, pre-operative sexual function scores, and remaining penile length, sexual function following partial penectomy was similar (i.e., the difference in scores was < 3 points between studies for most IIEF domains) across the four studies.

In addition to patient satisfaction, partner satisfaction is also an important outcome to consider when evaluating the impact of partial penectomy (or other surgical procedures) on sexual function. To this end, two studies [9, 18] administered the Erectile Dysfunction Inventory of Treatment Satisfaction (EDITS) questionnaire which includes both patient and partner scales. Although this questionnaire was not administered pre-operatively, thus precluding formal inclusion of these results in this systematic review, the key findings are discussed here briefly for the interested reader. Encouragingly, in the studies by both Wan et al. [18] (surveyed 6 months post operatively) and Sansalone et al. [9] (surveyed 3 months post-operatively), findings from the EDITS questionnaire suggest that both patients and their partners were highly satisfied with the outcome of their treatment, with average EDITS scores between 70–80 out of 100 (0 indicates extremely low satisfaction, 100 indicates extremely high satisfaction). Such information could be useful for helping inform patient and partner expectations, but should be interpreted cautiously given the absence of pre-operative data for comparison.

## Strengths and limitations

To the authors knowledge, this is the first study to systematically review the extant literature exploring the effects of partial penectomy for penile cancer on quantitative measures of sexual function. The study has several strengths, including the comprehensive search strategy which

was devised by an information specialist, adherence to the PRISMA guidelines, and the pre-registration of this study on the PROSPERO database to minimise bias. Several limitations are also worth highlighting. Firstly, there were only a small number of studies eligible for inclusion in this review. Although we identified several additional articles which reported measures of sexual function following partial penectomy, most of these investigations (e.g., [10, 19–29]) did not include a pre-surgery measure of sexual function specified in our prospectively registered inclusion/ exclusion criteria, which was deemed necessary to provide information on the pre- to post-surgery change in sexual function. A further limitation is that all of the studies included in this review were typically small, did not include a control group, and only included one follow up measure of sexual function, such that it was not possible to determine whether post-operative sexual function remains stable over time. In addition, there was a serious risk of bias in all included studies, which was primarily due to the inability to determine trends over time, lack of a control group, and lack of control for confounding variables in all studies. Moreover, only one investigation reported potential effect modifiers, and further studies are warranted which explore whether effects of partial penectomy on sexual function differs depending upon participant characteristics. In addition, none of the studies reported whether skin grafts were used for closure, which could impact upon length, cosmetic outcome and sexual function [46]. Additional details of the surgical procedure should be reported in future investigations, and studies may be warranted which contrast surgical outcomes (including sexual function) between primary closure and autologous reconstruction with a skin graft. Finally, most studies relied upon the IIEF questionnaire to evaluate the pre- to post-partial penectomy change in sexual function. This questionnaire has been criticised because it does not provide information on sexual stimulation via non-penetrative means or self-stimulation [47], and provides limited information on psychosexual background and partner relationship [9]. Future studies using a range of questionnaires which capture different aspects of sexual function pre- and post-partial penectomy are therefore warranted.

## Conclusions

In conclusion, the present systematic review demonstrates that, overall, there is typically a decrease in sexual function following partial penectomy for penile cancer. Nevertheless, many patients are still able to maintain satisfying sex lives post-operation, especially when greater penile lengths can be preserved, and in younger individuals with lower anxiety levels. Given the various limitations to the current body of evidence highlighted above, additional well-designed studies are warranted to provide further evidence on the effects of partial penectomy for penile cancer on sexual function.

## Supporting information

**S1 Checklist. PRISMA checklist.**
(DOCX)

**S1 File. Search strategy.**
(DOCX)

## Author Contributions

**Conceptualization:** Eleanor Whyte, Oliver M. Shannon, Alex Griffiths.

**Data curation:** Eleanor Whyte, Alexandra Sutcliffe, Oliver M. Shannon, Alex Griffiths.

**Formal analysis:** Eleanor Whyte, Alexandra Sutcliffe, Tom Clifford, Jamie Matu, Oliver M. Shannon, Alex Griffiths.

**Investigation:** Alex Griffiths.

**Methodology:** Eleanor Whyte, Philip Keegan, Tom Clifford, Jamie Matu, Oliver M. Shannon, Alex Griffiths.

**Project administration:** Eleanor Whyte.

**Supervision:** Philip Keegan, Alex Griffiths.

**Writing – original draft:** Eleanor Whyte, Oliver M. Shannon.

**Writing – review & editing:** Eleanor Whyte, Alexandra Sutcliffe, Philip Keegan, Tom Clifford, Jamie Matu, Oliver M. Shannon, Alex Griffiths.

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
