## [Decision Letter · Decision Letter 0]

23 Mar 2022

PONE-D-21-36652Effects of partial penectomy for penile cancer on sexual function: A systematic reviewPLOS ONE

Dear Dr. Griffiths,

Thank you for submitting your manuscript to PLOS ONE. After careful consideration, we feel that it has merit but does not fully meet PLOS ONE’s publication criteria as it currently stands. Therefore, we invite you to submit a revised version of the manuscript that addresses the points raised during the review process.

ACADEMIC EDITOR: The manuscript has been well written. Given 4 studies were identified and included in the systematic review. An updated search may help identify additional articles published after May 2021 and would enrich the content of the manuscript.

We look forward to receiving your revised manuscript.

Kind regards,

Monali S. Malvankar-Mehta, PhD

Academic Editor

PLOS ONE

Journal Requirements:

Additional Editor Comments (if provided):

The manuscript has been well written and requires minor corrections. The authors also address an important topic. The last search has been done till May 2021. I suggest updating the search and adding any additional papers published on the topic. This will help make the content of the manuscript richer.

Reviewers' comments:

Reviewer's Responses to Questions

**Comments to the Author**

1. Is the manuscript technically sound, and do the data support the conclusions?

Reviewer #1: Yes

2. Has the statistical analysis been performed appropriately and rigorously? 

Reviewer #1: Yes

3. Have the authors made all data underlying the findings in their manuscript fully available?

Reviewer #1: Yes

4. Is the manuscript presented in an intelligible fashion and written in standard English?

Reviewer #1: Yes

5. Review Comments to the Author

Reviewer #1: Dear Authors

Thank you for this opportunity to read your work. This is certainly a very interesting and relevant topic.

After considering your research approach, results, conclusions and limitations I have some comments regarding your study:

1. I do understand that you applied a narrative (descriptive) synthesis of the literature, but please refrain from using ambiguous statements like "was broadly comparable across the four studies".

2. You highlighted some very important limitations of the study, but I do appreciate the importance of your research.

Your research question(s) is clear, your results based on the applied method is supported and the manuscript is presented in an understandable and concise manner.

All the best in your future endeavors

6. PLOS authors have the option to publish the peer review history of their article (what does this mean?). If published, this will include your full peer review and any attached files.

Reviewer #1: No

---

## [Author Response · Author response to Decision Letter 0]

28 Mar 2022

REVIEWER #1 COMMENTS

Reviewer comment 1:

I do understand that you applied a narrative (descriptive) synthesis of the literature, but please refrain from using ambiguous statements like "was broadly comparable across the four studies".

Author response:

Thank you very much for this suggestion. We have re-read the text and made adjustments to avoid use of ambiguous language. In particular, for the sentence highlighted above around line 390-391, we have adjusted the text to the following, more specific language:

‘This suggests that despite the different surgical techniques, pre-operative sexual function scores, and remaining penile length, sexual function following partial penectomy was similar (i.e., the difference in scores was < 3 points between studies for most IIEF domains) across the four studies.’

Reviewer comment 2:

You highlighted some very important limitations of the study, but I do appreciate the importance of your research. Your research question(s) is clear, your results based on the applied method is supported and the manuscript is presented in an understandable and concise manner.

Author response:

Thank you very much, we are grateful for your feedback.

EDITORIAL COMMENTS

Editorial comment 1:

The manuscript has been well written. Given 4 studies were identified and included in the systematic review. An updated search may help identify additional articles published after May 2021 and would enrich the content of the manuscript.

Author response:

Thank you for your comment and for taking the time to appraise our manuscript. We have now re-run the searches (on 24th March, 2022) in the hope of identifying additional relevant research. We identified an additional 25 articles. Following removal of duplicates, we screened 14 titles and abstracts and 3 full-texts. Unfortunately, none of the articles was eligible for inclusion. We have updated our flowchart and the manuscript text accordingly.

---

## [Decision Letter · Decision Letter 1]

26 Jul 2022

PONE-D-21-36652R1Effects of partial penectomy for penile cancer on sexual function: A systematic reviewPLOS ONE

Dear Dr. Griffiths,

Thank you for submitting your manuscript to PLOS ONE. After careful consideration, we feel that it has merit but does not fully meet PLOS ONE’s publication criteria as it currently stands. Therefore, we invite you to submit a revised version of the manuscript that addresses the points raised during the review process.

The reviewers request to add a minor point in the discussion. Could you please rive the manuscript according to their suggestion?

We look forward to receiving your revised manuscript.

Kind regards,

Thomas Tischer

Staff Editor

PLOS ONE

Journal Requirements:

Reviewers' comments:

Reviewer's Responses to Questions

**Comments to the Author**

1. If the authors have adequately addressed your comments raised in a previous round of review and you feel that this manuscript is now acceptable for publication, you may indicate that here to bypass the “Comments to the Author” section, enter your conflict of interest statement in the “Confidential to Editor” section, and submit your "Accept" recommendation.

Reviewer #1: (No Response)

Reviewer #2: (No Response)

2. Is the manuscript technically sound, and do the data support the conclusions?

Reviewer #1: (No Response)

Reviewer #2: Yes

3. Has the statistical analysis been performed appropriately and rigorously? 

Reviewer #1: (No Response)

Reviewer #2: Yes

4. Have the authors made all data underlying the findings in their manuscript fully available?

Reviewer #1: (No Response)

Reviewer #2: Yes

5. Is the manuscript presented in an intelligible fashion and written in standard English?

Reviewer #1: (No Response)

Reviewer #2: Yes

6. Review Comments to the Author

Reviewer #1: (No Response)

Reviewer #2: Add discussion about Erectile Dysfunction Inventory of Treatment Satisfaction (EDITS) questionnaires.

7. PLOS authors have the option to publish the peer review history of their article (what does this mean?). If published, this will include your full peer review and any attached files.

Reviewer #1: No

Reviewer #2: No

---

## [Author Response · Author response to Decision Letter 1]

1 Sep 2022

REVIEWER #2 COMMENT

Reviewer comment 1:

Add discussion about Erectile Dysfunction Inventory of Treatment Satisfaction (EDITS) questionnaires.

Author response:

Thank you for this comment and for taking the time to review our manuscript. . We have now included a discussion about the EDITS questionnaire around lines 394-406.

---

## [Editor Report · Decision Letter 2]

7 Sep 2022

Effects of partial penectomy for penile cancer on sexual function: A systematic review

PONE-D-21-36652R2

Dear Dr. Griffiths,

We’re pleased to inform you that your manuscript has been judged scientifically suitable for publication and will be formally accepted for publication once it meets all outstanding technical requirements.

Kind regards,

Emily Chenette

Editor in Chief

PLOS ONE
---

## [Editor Report · Acceptance letter]

12 Sep 2022

PONE-D-21-36652R2 

Effects of partial penectomy for penile cancer on sexual function: A systematic review 

Dear Dr. Griffiths:

I'm pleased to inform you that your manuscript has been deemed suitable for publication in PLOS ONE. Congratulations! Your manuscript is now with our production department. 

Kind regards, 

on behalf of

Dr Emily Chenette 

Staff Editor

PLOS ONE